# Comprehensive Metagenomic Analysis of Veterinary Probiotics in Broiler Chickens

**DOI:** 10.3390/ani14131927

**Published:** 2024-06-29

**Authors:** Ádám Kerek, István László Román, Ábel Szabó, Márton Papp, Krisztián Bányai, Gábor Kardos, Eszter Kaszab, Krisztina Bali, László Makrai, Ákos Jerzsele

**Affiliations:** 1Department of Pharmacology and Toxicology, University of Veterinary Medicine, István utca 2, H-1078 Budapest, Hungary; roman.istvan.laszlo@student.univet.hu (I.L.R.); szabo.abel@student.univet.hu (Á.S.); banyai.krisztian@univet.hu (K.B.); jerzsele.akos@univet.hu (Á.J.); 2National Laboratory of Infectious Animal Diseases, Antimicrobial Resistance, Veterinary Public Health and Food Chain Safety, University of Veterinary Medicine Budapest, H-1078 Budapest, Hungary; papp.marton@univet.hu (M.P.); kg@med.unideb.hu (G.K.); kaszab.eszter@univet.hu (E.K.); bali.krisztina@univet.hu (K.B.); 3Centre for Bioinformatics, University of Veterinary Medicine, István utca 2, H-1078 Budapest, Hungary; 4Veterinary Medical Research Institute, Hungária krt. 21, H-1143 Budapest, Hungary; 5One Health Institute, University of Debrecen, Nagyerdei krt. 98, H-4032 Debrecen, Hungary; 6National Public Health Center, Albert Flórián út 2-6, H-1097 Budapest, Hungary; 7Department of Gerontology, Faculty of Health Sciences, University of Debrecen, Sóstói út 2-4, H-4400 Nyíregyháza, Hungary; 8Department of Microbiology and Infectious Diseases, University of Veterinary Medicine, István utca 2, H-1078 Budapest, Hungary; 9Autovakcina Kft., Szabadság sgrt. 57, H-1171 Budapest, Hungary; autovakcina@gmail.com

**Keywords:** probiotics, antimicrobial resistance, broilers, NGS, Ross308, chickens

## Abstract

**Simple Summary:**

Probiotics are beneficial bacteria used to improve health, but they might also carry genes that resist antibiotics, known as antimicrobial resistance genes (ARGs). These genes can transfer between bacteria, potentially leading to drug-resistant infections, a serious public health concern. In this study, we focused on probiotics used in broiler chickens—a key sector of the poultry industry due to their economic value and widespread consumption. Using advanced genetic sequencing techniques, we examined whether probiotics carry ARGs that could be transferred through mobile genetic elements like plasmids or phages. We also monitored changes in the gut bacteria and resistance genes in response to different probiotic treatments. Furthermore, we assessed how the genetic traits correlate with actual resistance to antibiotics by measuring the minimum inhibitory concentration (MIC) of drugs against the bacteria.

**Abstract:**

Probiotics are widely used in broiler chickens to support the gut microbiome, gut health, and to reduce the amount of antibiotics used. Despite their benefits, there is concern over their ability to carry and spread antimicrobial resistance genes (ARGs), posing a significant public health risk. This study utilized next-generation sequencing to investigate ARGs in probiotics approved for poultry, focusing on their potential to be transferred via mobile genetic elements such as plasmids and phages. We examined the gut microbiome and resistome changes in 60 broiler chickens over their rearing period, correlating these changes with different probiotic treatments. Specific resistance mechanisms against critically important antibiotics were identified, including genes related to fluoroquinolone resistance and peptide antibiotic resistance. We also found genes with significant relevance to public health (*aadK*, *AAC(6′)-Ii*) and multiple drug-resistance genes (*vmlR*, *ykkC*, *ykkD*, *msrC*, *clbA*, *eatAv*). Only one phage-encoded gene (*dfrA43*) was detected, with no evidence of plasmid or mobile genetic element transmission. Additionally, metagenomic analysis of fecal samples showed no significant changes corresponding to time or diet across groups. Our findings highlight the potential risks associated with the use of probiotics in poultry, particularly regarding the carriage of ARGs. It is crucial to conduct further research into the molecular genetics of probiotics to develop strategies that mitigate the risk of resistance gene transfer in agriculture, ensuring the safe and effective use of probiotics in animal husbandry.

## 1. Introduction

The term “probiotic” was first employed in 1974 to describe beneficial microorganisms that, when administered orally in suitable quantities, have positive effects on the host organism [1]. Probiotics are recognized for their immunomodulatory effects, facilitated by their interaction with intestinal epithelial cells and associated immune cells via receptors or direct adhesion [2,3]. Probiotic organisms compete with pathogens by adhering to the intestinal walls, effectively displacing harmful bacteria [4,5]. Additionally, probiotics can inhibit toxin production by pathogens, offering an antitoxin effect [6], and also produce bacteriocins such as lactic acid or hydrogen peroxide, which suppress the growth of certain pathogens [7]. Probiotics, along with other alternatives such as plant essential oils [8], antimicrobial peptides [9], or propolis [10,11,12,13], can be effective tools in preventing the spread of antimicrobial resistance (AMR) by partially or completely replacing the use of antibiotics.

Despite these advantages, probiotics also present risks. They can contribute to the development of various infections, such as fungemia caused by *Saccharomyces cerevisiae* or *Saccharomyces boulardii* [14,15,16,17,18], and there have been multiple reported cases of sepsis linked to species like *Lactobacillus*, *Bacillus subtilis*, or *Bifidobacterium breve* [19,20,21,22]. The potential excessive immunostimulatory effects of probiotics might also be problematic, as they can influence both innate and adaptive immune functions, triggering excessive immune responses that may lead to autoimmune diseases [23,24]. These contradictory experiences of probiotic use have led to their marketing solely as supplements, rather than as approved therapeutic agents [25,26]. A comprehensive study in 2011 concluded that although clinical trials have not proven increased health risks from probiotics, the available literature is insufficient to definitively confirm their safety [27].

The misuse of antibiotics and the related social and economic trends over the past few decades have significantly accelerated the selection and spread of resistant bacteria, with resistance-related deaths beginning to increase significantly [28]. Currently, around 700,000 deaths per year are linked to AMR, and estimates suggest this could rise to 10 million per year by 2050 if antibiotic use continues at the current rate and new therapies are not developed [29]. Notably, global human antibiotic use increased by 70% between 2000 and 2010 [30].

Because of their widespread use and the fact that probiotic ARG carriage may contribute to the spread of antimicrobial resistance [31], it is crucial to investigate the role of probiotics in the transfer of antimicrobial resistance genes (ARGs). Probiotics must not become a source of specific resistance genes [32], but this issue has not yet been agreed upon globally. Key organizations, including the Biosafety Assessment of Probiotics used for Human Consumption (PROSAFE), the Assessment and Critical Evaluation of Antibiotic Resistance Transferability in Food Chain (ACE-ART) project, and the Joint International Organization for Standardization-International Dairy Federation (ISO-IDF), are addressing this topic [33,34,35]. Metagenomic analyses have shown that environmental factors and probiotic feeding can influence ARG levels in feces. For instance, one study found that feeding *Bacillus coagulans* as a probiotic strain significantly increased the appearance of aminoglycoside ARGs in the feces of laying hens exposed to lead [36]. To address the current situation, it is essential to establish up-to-date regulations for probiotics, including antimicrobial resistance (AMR) testing. Additionally, guidelines for their proper use and disposal should be introduced [37]. New acute and long-term risks associated with probiotics have emerged, highlighting the need to update safety recommendations for these products [38].

The most commonly used probiotic strains belong to the genera *Lactobacillus*, *Lactococcus*, *Bacillus*, *Enterococcus*, and *Bifidobacterium* [39,40]. However, *Enterococcus* species, which harbor numerous ARGs, may not be the most suitable probiotic strains due to their potential for resistance gene transfer [41,42,43] and the fact that their health benefits have been clinically demonstrated in a limited number of cases [44,45].

Antibacterial resistance can be advantageous for probiotic strains to survive antibiotic treatment. However, these genes can be transmitted vertically or horizontally between bacteria. The gastrointestinal tract of higher organisms provides favorable conditions for gene transfer. In contrast, this process does not pose a threat in the case of yeasts, as there is no gene transfer between yeasts and bacteria, making their use considered safe [46]. It is important to distinguish between intrinsic antimicrobial resistance, which is not horizontally transferable, and acquired antimicrobial resistance, which can be transferred through various routes in probiotic strains. For instance, *Lactobacillus* species exhibit intrinsic resistance to several antibiotic groups, including aminoglycosides, glycopeptides, nucleic acid synthesis inhibitors, and folate synthesis inhibitors [47]. Generally, they are sensitive to penicillins and β-lactam inhibitors but show reduced sensitivity to cephalosporins. Nonetheless, several studies have reported resistance to penicillin G in various isolates [48,49,50]. Current research findings are often contradictory, necessitating more extensive and in-depth studies. Some research indicates that feeding preparations containing *Bacillus licheniformis* with different bacterial genomes affects the amount of antimicrobial resistance genes (ARGs) in the cecal and colonic contents of chickens [51]. In contrast, other studies suggest that a significant proportion of ARGs in the gut microbiome are acquired ARGs [52]. Metagenomic studies have demonstrated that feeding probiotics, in combination with various environmental factors, can influence the quantity of ARGs in the gut [36].

Our study aims to assess the safety of probiotics in poultry, focusing on the potential transfer of antimicrobial resistance genes within the gut microbiome of broiler chickens, thereby contributing to safer long-term agricultural practices.

## 2. Materials and Methods

### 2.1. Conditions of the Animal Experiment

The animal experiment was conducted at the Animal House of the Department of Pharmacology and Toxicology, University of Veterinary Medicine, Budapest. The animal experiments were approved by the ethical committee of the Veterinary University of Budapest (Licence No. ÁTET/EA/23/2022), the result of Act No. 28 of 1998.3. § (9), as well as 40/2013. 1. § (4), (c), and especially points (f) of the Government Decree; they are not classified as examination adversely affecting animal welfare, so no ethical permit is required. The experiment involved mixing five different probiotic products with feed and giving them to 60 day-old Ross308 broiler chicks sourced from Pipi-Tér Bt. (Bábolna TETRA Ltd., Budapest, Hungary). Before arrival, chicks were vaccinated with Hatchback AVINEW, Hatchback IBD120, and RISMAVAC + CA126 at the hatchery.

The chicks were housed in cages designed to meet animal welfare standards, with environmental conditions adjusted according to their age (Appendix A). Each chick was uniquely identified by a numbered tag attached to its foot, facilitating the tracking of weekly weight gain and the collection of fecal samples. The chicks were fed ad libitum with broiler starter (0–14 days), broiler grower (15–30 days), and broiler finishing (31–42 days) feed (Appendix A) and had access to fresh drinking water daily.

### 2.2. The Probiotic Products Tested and Their Dosage

Five different probiotic products, authorized for use in poultry feed, were selected (Table 1). For ARG analysis, 1 g samples of each product were sent for genome analysis using next-generation sequencing.

The selection of probiotic products was based not only on their relevance to poultry research but also on their widespread use and the antimicrobial gene profiles obtained through next-generation sequencing (NGS). This selection is significant not only for animal health but also for public health. The products were specifically chosen to include strains with the highest risk and the most ARGs to thoroughly investigate their impact and behavior. NGS testing was conducted prior to selection to identify those strains that carry significant ARGs, allowing us to study their potential for ARG transfer and the implications for both animal and human health. This approach ensures a comprehensive understanding of the risks associated with these probiotic strains.

A total of 60 animals were divided into six groups, with 10 animals per group. The first group, serving as the control, was not administered probiotics. The remaining five groups were given the manufacturers’ recommended dosages of the selected probiotic products from day 5 until the end of rearing on day 42. The dosages were as follows: BioPlus YC was administered at 4 g/10 kg of feed, Agroferm M+C at 7 g/10 kg of feed; Ecobiol WX at 1 g/10 kg of feed; Gastroferm M+C at 7 g/10 kg of feed; and Fecinor Soluble Plus at 1 g/10 kg of feed. The probiotics were uniformly mixed into the feed using a planetary feed mixer, with a homogenization time of two hours per 10 kg of feed.

### 2.3. Measurement of Animal Body Weight and Sampling

Animal body weight was measured individually using a scale once a week. Cloaca swab samples for sequencing were collected twice during the rearing period: at day-old age before the treatment, and after the treatment. Fecal samples were taken directly from the cloaca using a sterile AMIES sampler (Biolab Zrt., Budapest, Hungary), individually from each animal, allowing for the comparison of sequencing results by individually marking the animals. The initial sampling occurred on day 4, followed by a final sampling at the end of the rearing period on day 42. In the case of the group fed Ecobiol WX, animals were sampled twice at each time point using sterile swabs. For all sampled animals, 3 mL of sterile phosphate buffer solution was measured into sampling tubes in a sterile cabin prior to collection, and fecal samples were subsequently stored at −80 °C until sequencing. For the Ecobiol WX group, 3 mL of tryptone soy broth (TSB) was also measured into the second sampling tube under a laminar box for further enrichment culture.

The differences in body weight gain among the groups were compared with the control group for each product and statistically analyzed using two-way ANOVA [53] with R version 4.1.0 [54].

### 2.4. Analysis of Phenotypic Expression

The phenotypic expression of antimicrobial resistance was assessed by determining the minimum inhibitory concentration (MIC) of isolated probiotic strains against specific antibiotics, which have animal and public health significance. An inoculating loop of isolated bacteria was placed in a sterile tube containing 3 mL of Mueller–Hinton broth (MHB) and incubated at 37 °C for 18–24 h the day before testing. Stock solutions of test substances (Merck KGaA, Darmstadt, Germany) were prepared according to Clinical Laboratory Standards Institute (CLSI) methodology [55]. In the 96-well microtiter plate (VWR International, LLC, Debrecen, Hungary), all columns except the first were filled with 90 µL of MHB, and the first column received 180 µL of the stock solution. A two-fold dilution series was prepared by transferring 90 µL from the first column into the second column, thoroughly mixing 3–4 times. This process was continued to column 10, where the excess 90 µL was discarded.

Next, the bacterial suspension, adjusted to a 0.5 McFarland standard, was inoculated onto the plates [55]. From column 11 of the working plates, containing two dilution lines and working backwards, 10 µL of bacterial suspension was added to each well. Column 11 served as a positive control, receiving both tap water and bacterial suspension but no active substance, while column 12 acted as a negative control, containing only tap water, with no bacteria or active substance.

Evaluation of the MICs was performed using an SWIN automatic MIC reader and the VIZION system (CheBio Development Ltd., Budapest, Hungary). Breakpoints for each antimicrobial agent were determined based on CLSI and the European Committee on Antimicrobial Susceptibility Testing (EUCAST) guidelines. *Escherichia coli* (ATCC 25922) was used as the reference isolate.

Tiamulin and florfenicol were selected based on resistance-encoding genes of public health significance identified in the sequencing of Ecobiol WX. The two antibiotics were enriched in broth containing half the MIC concentration (32 µg/mL for tiamulin and 2 µg/mL for florfenicol). Samples were separately taken from the treated groups to select for resistant microbial strains. The samples were then kept in the enrichment broth at 37 °C for 18–24 h.

### 2.5. Next Generation Sequencing (NGS)

Probiotic products were sequenced using an Illumina NextSeq 500 system [56]. DNA was extracted from 1 g of each product using the QIAmp DNA kit (Qiagen, Hilden, Germany) following the manufacturer’s protocol. Fecal samples were sequenced on a NovaSeq PE150 platform [57], with DNA extractions also performed using the QIAmp DNA kit, according to the manufacturer’s protocol. For the bead shaking required to liberate bacterial genetic material, a Qiagen TissueLyzer LT (Qiagen GmBH, Hilden, Germany) was used at 50 Hz for 5 min. The extracted nucleic acid was stored at −20 °C until use. Illumina’s sequencing procedure utilizes a “paired-end” technique where strands are anchored with oligonucleotides in a bridge amplification. The complementary strand is then synthesized and bridged, followed by the removal of the reverse strand. Fluorescently labeled nucleotides are read during sequencing, allowing for high accuracy in the identification of the genetic sequences [58,59].

In total, 28 pooled samples were sequenced—four from each group. For each sequenced sample, five samples from each group were combined to produce two samples for sequencing, at both the beginning and end of the experiment. For next-generation sequencing, DNA libraries were prepared using the Illumina^®^ Nextera XT DNA Library Preparation Kit (Illumina, San Diego, CA, USA). Indexes from the Nextera XT Index Kit v2 Set A (Illumina, San Diego, CA, USA) were used to uniquely label DNA fragments. First, nucleic acid samples were diluted to a concentration of 0.2 ng/µL in a final volume of 2.5 µL and mixed with 5 µL of Tagment DNA buffer and 2.5 µL of Amplicon Tagment Mix reagent for the tagmentation reaction. The reaction mixture was incubated at 55 °C for 6 min in an Eppendorf Mastercycler nexus GX2 (Eppendorf SE, Hamburg, Germany) and then cooled to 10 °C. Subsequently, 2.5 µL of Neutralize Tagment buffer was immediately added and incubated for 5 min at room temperature.

For DNA library preparation, 7.5 µL of Nextera PCR Master Mix was combined with 2.5 µL each of i5 and i7 index primers and added to the tagmented DNA sample for PCR amplification. The PCR program consisted of an initial denaturation at 95 °C for 30 s, followed by 12 cycles of 95 °C for 10 s, 55 °C for 30 s, and 72 °C for 30 s, with a final elongation step at 72 °C for 5 min, after which the samples were cooled to 10 °C. The resulting indexed DNA library was purified using a Gel/PCR DNA Fragments Extraction Kit (Geneaid Biotech, Xinpei, Taiwan) following the column purification protocol. The purified libraries were then quantified fluorometrically using a Qubit^®^ dsDNA HS Assay Kit (Thermo Fisher Scientific, Waltham, MA, USA). Finally, DNA libraries with individual adapters were diluted to the appropriate concentration and mixed for sequencing.

### 2.6. Bioinformatics Data Analysis

Quality control of the raw sequence data was conducted using FastQC v0.11.9 [60], and Fastp v0.23.2-3 [61] and Bloocoo v1.0.7 [62] were used to filter out and correct possible adapter sequences, unbalanced base pair distributions, and poor-quality regions. Invalid sequences were filtered out using TrimGalore v0.6.6 [63]. The reads were then assembled into longer sequences (contigs) using MEGAHIT v1.2.9 [64]. The results from the two assemblies were then combined to produce an even-higher-quality draft genome using GAM-NGS v1.1b [65]. Contigs underwent quality control assessment using QUAST software [66]. To exclude fragments of the domestic fowl genome contained within the gut contents, read sequences were aligned to the reference genome of *Gallus gallus* (NCBI identifier: GRCg6a) using Bowtie2 software [67]. Taxonomic classification of sequences was performed using Kraken2 [68] software against the NCBI nucleotide database [69]. Analysis was carried out in an R-environment [54], utilizing phyloseq [70] and microbiome [71] packages. Bacterial origin sequences were constructed using metaSPAdes [72] software. Open reading frames (ORFs) from the resulting contigs were determined using Prodigal v2.6.3 [73]. ARG identification among ORFs was conducted using the Resistance Gene Identifier (RGI) v5.1.0 in comparison with The Comprehensive Antibiotic Resistance Database (CARD) [74]; only genes meeting the STRICT threshold defined by CARD and showing at least 95% sequence identity and coverage were considered. During the taxonomic analyses, the Shannon diversity index, Mann–Whitney test, and NMDS ordination analysis were performed using R version 4.1.0. [54].

The identified resistance genes were analyzed for potential mobility using MobileElementFinder v1.0.3 [75], considering only ARGs located within the longest transposon distance defined for a given microorganism in the database. The plasmid origin of contigs was investigated using PlasFlow v1.1 software [76], and phage encoding of the genes was assessed using VirSorter v2.2.2 [77] software.

## 3. Results

### 3.1. Animal Body Weight Gain

Weight gain for each group was calculated on a weekly basis, as shown in Figure 1. Body mass increased steadily throughout the duration of the trial. Statistical analysis revealed no significant differences between the control group (Group I) and the other treated groups (Appendix A). However, Group II (BioPlus YC) and Group IV (Ecobiol WX) demonstrated notably higher (but not significant) average weight gains compared to the control. The standard weight gain expected for the Ross308 breed was consistently met or exceeded by the groups receiving each formulation. Notably, by week 4 of life, weight gains in Group II had increased by 5%, in Group IV by 6%, in Group V (Gastroferm M+C) by 3%, and in Group VI (Fecinor Soluble Plus) by 3%. At week 5, Groups II and IV continued to show enhanced weight gains of 3% over the control.

No statistically significant differences were observed between the final body weights of any of the experimental groups and the control group (Group I). However, from a commercial perspective, measurements showing a positive mean weight difference of at least 5% compared to the control group were deemed relevant. This criterion was met in 42% of the cases for group averages. Notably, Group IV consistently exhibited a weight difference greater than 5% compared to the control group (exception in week 5). To achieve statistically significant results, it is recommended that the sample size be increased in future studies.

### 3.2. Results of NGS Regarding the Probiotic Products

The antimicrobial resistance genes (ARGs) identified in each probiotic product following sequencing are detailed in Table 2. A key mechanism of resistance, enzymatic inactivation, was observed in several samples. Notably, the *aadk* gene, responsible for aminoglycoside resistance and originating from *Bacillus* spp., was identified in BioPlus YC. The *AAC(6’)-Ii* gene, another aminoglycoside resistance gene but of *Enterococcus* spp. origin, was detected in Agroferm M+C, Gastroferm M+C, and Fecinor Soluble Plus.

Additionally, BioPlus YC was found to contain the *mphk* gene, which is associated with macrolide antibiotic resistance. Among the target mutations, the *mprF* gene warrants special attention, as it has been identified in BioPlus YC and linked to peptide antibiotic resistance. Furthermore, the diaminopyrimidine resistance gene *dfrA43* was identified in Ecobiol WS. Despite being encoded by a dsDNA phage from *Bacillus* spp., it exhibited relatively low coverage (82.56%) and identity (35.75%), highlighting the complexity of accurately assessing gene origins and functions.

Most efflux-pump-related genes, including those responsible for fluoroquinolone resistance (*blt*, *bmr*) and peptide antibiotic resistance (*bcrA*, *bcrB*), were identified in BioPlus YC. Additionally, the *efmA* gene, which codes for an efflux pump conferring resistance to both macrolides and fluoroquinolones, was found in both Agroferm M+C and Gastroferm M+C. The presence of the *ykkC* and *ykkD* genes in BioPlus YC, which are associated with multi-resistance to aminoglycosides, tetracyclines, and phenicols, is also notable. Importantly, no mobile genetic elements (MGEs) were identified, and no antimicrobial resistance genes (ARGs) encoded on plasmids were detected.

### 3.3. Phenotypic Analysis of Probiotic Strains

In the *Bacillus licheniformis* and *Bacillus subtilis* strains from BioPlus YC, no antimicrobial resistance genes (ARGs) related to penicillins were identified, indicating phenotypic sensitivity. Resistance to gentamicin, demonstrated by high minimum inhibitory concentrations (MICs) of 8 and 16 µg/mL, is attributable to the expression of the *aadK* gene (enzymatic inactivation) and the *ykkC* and *ykkD* genes (efflux pumps). The high MIC values for oxytetracycline (8 µg/mL) in *Bacillus licheniformis* and for doxycycline (32 µg/mL) in *Bacillus subtilis* suggest resistance mechanisms involving the *vmlR* gene (target mutation) or the *ykkC* and *ykkD* genes (efflux pumps). Clindamycin resistance (32 µg/mL) in the former strain could be due to the *vmlR* or *ermD* genes (target mutations) or the *lmrB* gene (efflux pump). In both strains, high tiamulin MICs (64 µg/mL) were linked to the *vmlR* gene (target mutation).

For Agroferm M+C, all identified ARGs are of *Enterococcus* origin, with inherent resistance to cephalosporins indicated by a ceftriaxone MIC of 16 µg/mL, although generally showing sensitivity to penicillins. Gentamicin resistance (32 µg/mL) may be mediated by the *AAC(6’)-Ii* gene (enzymatic inactivation), and florfenicol resistance (8 µg/mL) may involve the *msrC* or *eatAv* genes (target mutations). These genes are also implicated in the high tiamulin MICs (64 µg/mL), a resistance pattern observed similarly in *Lactobacillus plantarum*. MIC values for probiotic strains are detailed in Table 3 and Table 4.

For Ecobiol WX, resistance to tiamulin (64 µg/mL) was associated with the *clbA* gene (target mutation). In *Lactobacillus plantarum*, *Pediococcus acidilactici*, and *Enterococcus faecium*, gentamicin resistance (32 µg/mL) may be attributed to the *AAC(6’)-Ii* gene (enzymatic inactivation). In *Pediococcus acidilactici*, resistance to oxytetracycline (32 µg/mL) and doxycycline (16 µg/mL) could be caused by the *msrC* or *eatAv* genes (target mutation). Additionally, all three strains exhibited resistance to florfenicol (8 µg/mL) and tiamulin (>64 µg/mL), potentially due to activation of the *msrC* or *eatAv* genes (target mutation).

For Fecinor Soluble Plus, observed aminoglycoside resistance (32 µg/mL) could be linked to the *AAC(6’)-Ii* gene (enzymatic inactivation). Resistance to florfenicol (8 µg/mL) and tiamulin (64 µg/mL) may also result from activation of the *msrC* or *eatAv* genes (target mutation) [78]. The elevated MIC values for other active substances suggest inherent (*ab ovo*) resistance. However, to conclusively determine which phenotypic resistance manifestations result from gene activation, further transcriptomic studies are required.

### 3.4. Gut Microbiome Composition and Antimicrobial Resistance Gene Expression

An average of 36,549,401 reads (minimum: 16,105,059; median: 36,387,144; maximum: 59,792,673) were obtained from the bacterial kingdom across the samples, with 29,277,955 reads (minimum: 14,359,056; median: 26,649,405; maximum: 54,262,552) classified at the genus level and 25,885,945 reads (minimum: 7,610,791; median: 24,713,784; maximum: 51,592,030) at the species level. The core bacteriome, defined as genomes present in at least 10% of the samples with a minimum abundance of 1%, was analyzed.

At the strain level, Firmicutes and Proteobacteria were dominant. In the control group, the proportion of Firmicutes increased from 28.5% to 59%, while Proteobacteria decreased from 66% to 31% by the end of the experiment. In contrast, when fed BioPlus, the proportions of these phyla remained stable (from 63.5% to 66% for Firmicutes and from 26.5% to 25% for Proteobacteria). However, in groups fed other probiotics, the initial dominance of Firmicutes significantly decreased by the end of rearing, with corresponding increases in Proteobacteria.

The proportion of Bacteroidetes rose markedly in the control group (from 0.04% to 10.03%) but remained below 1% in all other groups. The addition of tiamulin and florfenicol noticeably altered the microbiome in the Ecobiol-fed group, drastically reducing the proportion of Firmicutes, including Gram-positive bacteria (from 81% to 1.3% at the beginning and from 64% to 4% at the end of rearing), while dramatically increasing the proportion of Proteobacteria, including Gram-negative bacteria (from 6% to 97% and from 25.5% to 93.5%).

At the genus level, *Lactobacillus* was predominant in most samples. In the control group, the proportion of *Lactobacillus* increased from 18.5% to 34%. Similarly, in the group fed Gastroferm, *Lactobacillus* levels rose from 19% to 50%. In contrast, the other groups showed a decrease. The proportion of *Enterococcus*, the second dominant genus, declined slightly in the control group (from 7.5% to 6.5%) and also in all the other groups, except for BioPlus, which saw an increase (from 4% to 5.5%). At the genus level, Shannon diversity distributions by group and sampling time did not show significant results, nor did the Mann–Whitney test reveal significant differences in gene composition before and after feeding probiotics (Appendix A). Likewise, no significant difference in the number of genes observed was found by the Mann–Whitney test (Appendix A). NMDS ordination based on Bray–Curtis distances between samples is shown before feeding in Appendix A and after feeding in Appendix A. The abundance of the core bacteriome (genes present in at least 10% of the samples and at a minimum abundance of 1%) in the pre- and post-feeding samples is summarized in Appendix A.

Regarding the genus *Escherichia* from the Proteobacteria phylum, increases were noted in the Agroferm (from 4% to 5.5%) and Fecinor groups (from 3.5% to 5.5%), while decreases were observed in all other groups. The proportion of *Klebsiella* significantly increased across all groups: control (from 0.1% to 9%); BioPlus (from 0.05% to 11%); Agroferm (from 0.02% to 8.5%); Ecobiol (from 0.06% to 17.5%); Gastroferm (from 0.02% to 8%); and Fecinor (from 1.01% to 12%).

In enriched samples from the Ecobiol group, a dominant selection of Gram-negative bacteria was observed at the genus level. The percentage of *Escherichia* increased from 2.5% to 21% and *Proteus* from 0.09% to 10%, as illustrated in Figure 2.

The variation in species composition of each probiotic-treated group over time is illustrated in Figure 3. At a few days old, the samples were dominated by *Escherichia coli*, *Lactobacillus johnsonii*, *Enterococcus* species, and *Ligilactobacillus salivarius*. By day 42, there was a significant decrease in the proportion of *Enterococcus* species compared to the first day, while the proportion of *Klebsiella pneumoniae* increased significantly. *Lactobacillus crispatus*, *Lactobacillus helveticus*, and *Lactobacillus amylovorus* species became dominant after treatment with each probiotic, except for *Lactobacillus johnsonii*, which showed a significant decrease in all cases. Shannon’s species-level analysis of diversity between groups and Mann–Whitney tests of paired samples within groups revealed no significant differences before and after feeding probiotics (Appendix A). The distribution of the number of species observed, controlled for sampling time and feeding, in samples paired within groups using Mann–Whitney tests showed no significant difference between samples taken at the beginning and end of the period (Appendix A). NMDS ordination based on Bray–Curtis distances between samples at the species level before feeding is shown in Appendix A and after feeding in Appendix A. Abundances of the core bacteriome, defined as bacterial species present in at least 10% of samples at a minimum abundance of 1%, are displayed in Appendix A.

The changes in the gut microbiome at the strain level due to enrichment are illustrated in Figure 4. This figure clearly shows the selection of Gram-negative bacteria, while the prevalence of Gram-positive bacteria is only minimally reduced. The abundances of the core bacteriome per sample, defined as at least 1% of the genes being present in at least 10% of the samples, are shown in Appendix A. The same data at the species level are displayed in Appendix A.

A total of 172 antimicrobial resistance genes (ARGs) were identified across all samples. The ARGs were categorized by antibiotic drug class and mechanism of resistance. Analysis revealed no significant differences in the distribution of these genes between the control group and the groups fed each probiotic product, neither at the beginning nor at the end of the rearing period (Appendix A).

The *AAC(6’)-Ii* gene (associated with resistance to aminoglycosides via enzymatic inactivation), the *eatAv* gene (associated with resistance to pleuromutilins via target protection), and the *efmA* gene (associated with resistance to macrolides and fluoroquinolones via efflux pumping) were present in initial samples taken on day 4. These genes were not detected in samples collected at the end of the rearing period, with the exception of the Gastroferm group, where these genes were inherently carried by the probiotic used.

Additionally, it is concerning that genes such as *SHV-38*, *TEM-1*, and *TEM-163*, which are responsible for the production of extended-spectrum β-lactamases (ESBLs), were identified in several samples taken from 4-day old chicks.

## 4. Discussion

In this study, the effects of five different probiotic products on broiler chickens over a 42-day rearing period were evaluated. Although no statistically significant differences in weight gain were observed between the control group and the probiotic-fed groups (*p* > 0.05), specific formulations such as BioPlus YC and Ecobiol WX demonstrated a commercially relevant increase in weight gain by the fourth week (+3–6%). Notably, only BioPlus YC and Ecobiol WX maintained this increase by the fifth week (+3%). The Ecobiol group consistently showed significantly more weight gain compared to the control throughout the rearing period.

These findings partially align with Xu et al., who observed significant increases in feed intake and weight gain with Bacillus subtilis and Bacillus licheniformis (10^9^ CFU/g) in a similar study involving 360 roosters [79]. However, in our study, BioPlus, containing the same probiotics, did not yield a statistically significant difference from the control group (*p* > 0.05). This contrasts with the results from Ma et al., who reported significant weight gains on day 42 with *Bacillus subtilis* compared to the control group (*p* = 0.022) [80], whereas our findings showed no significant difference (*p* = 0.32).

Further comparisons reveal a mixed response to probiotic supplementation in poultry. Trela et al. reported significant weight gains in the early stages of rearing with *Bacillus licheniformis*, but no differences in later stages [81]. Similarly, Wang et al. noted a significant weight gain with *Lactobacillus plantarum* only at day 21, but not after [82]. Conversely, studies by Reuben et al. with *Pediococcus acidilactici* [83], Wu et al. with *Enterococcus faecium* [84], and Han et al. with *Enterococcus faecalis* saw noticeable weight gain only at day 42 [85] These results indicate significant probiotic-related weight gains occur only during specific phases of the growth period, and underscore the variability in probiotic efficacy across different strains and feeding schedules.

Our study highlights the dynamic changes in the gut microbiome of broiler chickens, dominated by shifts in the populations of Firmicutes and Proteobacteria, over a 42-day rearing period. Specifically, in the control group, the amount of Firmicutes increased significantly from 28.5% to 59%, while the proportion of Proteobacteria decreased from 66% to 31%. Interestingly, in chickens fed BioPlus, the proportion of these phyla remained stable (Firmicutes: 63.5% to 66%; Proteobacteria: 26.5% to 25%), suggesting that this probiotic might stabilize populations of these key microbial groups. Furthermore, although the effects were not statistically significant, the weight gain observed in the group fed this diet was consistently better across all measurement points.

Conversely, other probiotic treatments initially increased the proportion of Firmicutes, which subsequently decreased towards the end of the rearing period. This change was accompanied by an inverse trend in Proteobacteria. This fluctuation contrasts with findings by Ma et al., who reported a similar dominance of Firmicutes and Proteobacteria but no significant changes in microbiome diversity between control and treated groups, particularly when Bacillus subtilis was administered [80]. Han et al. also found no change in microbial diversity with *Enterococcus faecalis* treatment [85], and Trela et al. observed significant differences only in fecal samples from the jejunum [81].

In our analysis, only the control group exhibited a notable increase in Bacteroidetes, reaching 10.03% by the end of the period, while all other groups maintained a rate below 1%. This observation aligns with Ma et al., who reported a significant reduction in Bacteroidetes with probiotic treatment [80]. Similarly, Gao et al. found comparable levels of Firmicutes and Proteobacteria in broilers, across different treatments, underscoring the potential for probiotics to modify the gut flora composition [86].

In our study, the proportion of the *Lactobacillus* genus in the control group increased significantly from 18.5% to 34%. Among the treated groups, only chickens receiving Gastroferm exhibited a substantial increase, from 19% to 50%, whereas all other groups experienced a decrease. This pattern highlights the specific impact of Gastroferm on promoting *Lactobacillus* growth, aligning with findings by Sureshkumar et al., who reported a prevalence of 22.82% [87], and Gao et al., who noted a 16.69% prevalence in similar settings at the end of the study period [86]. Conversely, the *Enterococcus* genus displayed a slight decrease in the control group, from 7.5% to 6.5%, while increasing slightly in the BioPlus-treated group from 4% to 5.5%. This indicates that BioPlus may support *Enterococcus* growth. Furthermore, our study found that the *Escherichia* genus saw increases in groups treated with Agroferm and Fecinor, from 4% to 5.5% and 3.5% to 5.5%, respectively. In contrast, all other groups showed a decrease in this genus, suggesting that Agroferm and Fecinor may selectively promote growth of *Escherichia* species. Remarkably, we observed a significant increase in the *Klebsiella* genus across all groups, with percentages rising considerably during the rearing period (Control: 0.1% to 9%; BioPlus: 0.05% to 11%; Agroferm: 0.02% to 8.5%; Ecobiol: 0.06% to 17.5%; Gastroferm: 0.02% to 8%; Fecinor: 1.01% to 12%). These findings are consistent with those reported by Khan et al., who also observed a notable increase in *Klebsiella* following probiotic treatment [88]. This raises potential concerns about the pathogenic risks associated with this genus. Moreover, compared to other studies reporting higher prevalences for *Enterococcus*, *Blautia*, and *Clostridium*, our results indicated lower average prevalences (0.41% and 0.84%, respectively) for the latter two genera, which could reflect variations in dietary composition, probiotic formulations, or baseline microbiota [87].

Our analysis of the ARG pool within the tested probiotic products revealed a diverse array of resistance genes. Notably, the BioPlus product contained a significant number of genes, including the fluoroquinolone resistance genes *blt* and *bmr*. The *blt* gene, identified in *Bacillus subtilis* by Ahmed et al. [89], and the *bmr* gene, described by Neyfakh et al. and Klyachkyo et al. [90,91], are critical markers of resistance in this context. Additionally, the *bcrA* and *bcrB* genes, associated with peptide antibiotic resistance and described in *Bacillus licheniformis* by Podlesek et al. [92], were also present.

The presence of the *ykkC* and *ykkD* genes, which confer multidrug resistance to aminoglycosides, tetracycline, and phenicol, further underscores the broad-spectrum resistance potential in these probiotic strains, as detailed in *Bacillus subtilis* by Jack et al. [93]. The *aadk* gene, a key player in enzymatic inactivation leading to aminoglycoside resistance, was noted by Argerso et al. in *Bacillus licheniformis* [94]. Among efflux pump-related genes, the *mphk* gene implicated in macrolide resistance was shown in *Bacillus subtilis* by Pawlowski et al. [95], and the *bcrC* gene, linked to peptide antibiotic resistance as a target mutation, was also highlighted [92].

Target mutations such as those caused by the *mprF* gene, which can lead to peptide antibiotic resistance, have been identified in *Bacillus subtilis* [96,97,98]. In the Agroferm and Gastroferm products, the *efmA* gene was observed encoding an MFS-type efflux pump that contributes to resistance against macrolides and fluoroquinolones, as identified in *Enterococcus faecium* by Urshev et al. Furthermore, the *AAC(6’)-Ii* gene, responsible for aminoglycoside resistance, was found in Agroferm, Gastroferm, and Fecinor Soluble Plus, indicating a notable prevalence of this resistance mechanism in *Enterococcus faecium* [99].

In our study, the differences in antimicrobial resistance genes (ARGs) within the fecal microbiomes of treated and control groups were not statistically significant over the 42-day period. This observation aligns with findings by Chen et al., who reported a reduction in the variety of ARG (Fisher’s alpha) in chickens fed probiotics [51]. However, the persistence of certain ARGs warrants special attention due to their implications for public health.

Notably, the *AAC(6’)-Ii* gene, which is involved in the enzymatic inactivation of aminoglycosides, was lost from all treatment groups, except those fed Agroferm, by the end of the rearing period. This gene has been extensively studied in *Enterococcus faecium* and is recognized for its clinical significance [99]. The persistence of this gene in the Agroferm group suggests a selective pressure maintained by the probiotic formulation that may affect the dissemination of resistance traits.

Additionally, the detection of the *TEM-1* gene in all groups and the *TEM-163* gene in one group of 4-day old chicks, along with the emergence of the *SHV-38* gene, raises concerns. These genes are associated with extended-spectrum beta-lactamase (ESBL) production, a major resistance mechanism that complicates treatment of bacterial infections. Hassen et al. found the *TEM-1* gene in 25.8% of chicken feces and meat samples, highlighting its prevalence in poultry environments [100]. The *SHV-38* gene, typically identified in Klebsiella pneumoniae, has not been previously reported in chicken feces [101]. Similarly, the presence of the *TEM-163* gene, also linked to ESBL production and identified in *E. coli*, underscores the potential risk of resistance transfer through the food chain [102].

These findings indicate that while probiotics can influence the ARG profile of the gut microbiome, the emergence and persistence of significant resistance genes such as *AAC(6’)-Ii*, *TEM-1*, *SHV-38*, and *TEM-163* require careful monitoring. The potential for these ARGs to spread within agricultural settings and into clinical environments calls for enhanced surveillance and targeted interventions to mitigate the risks associated with antibiotic resistance in poultry production. Further studies should explore the mechanisms by which different probiotics influence microbial populations and their long-term impacts on poultry health and productivity.

## 5. Conclusions

The carriage of antimicrobial resistance genes (ARGs) in probiotics approved for poultry represents a critical yet under-researched area posing significant public health threats. Our study utilized next-generation sequencing to examine the ARG profiles in various probiotic products and analyzed changes in the fecal resistome and microbiome of broiler chickens, correlating these with phenotypic expressions of resistance through minimum inhibitory concentration (MIC) values.

We identified several genes responsible for resistance to critically important antibiotics, including fluoroquinolone efflux pumps and peptide antibiotic target mutations. Notably, genes facilitating aminoglycoside enzymatic inactivation, such as aadK and AAC(6’)-Ii, were of significant public health relevance. Multiple multidrug resistance genes, such as vmlR, ykkC, ykkD, msrC, clbA, and eatAv, were also found. However, our findings indicated the absence of mobile genetic element (MGE) or plasmid-encoded genes, with only one phage-encoded gene (dfrA43) detected. No evidence of direct gene transfer from probiotics to the gut microbiome was observed.

Phenotypic resistance testing through MIC scoring revealed specific resistance-associated genes in the examined strains, but no significant differences were noted in Shannon diversity distribution and Mann–Whitney tests of the fecal metagenome at the genus level over time and across different probiotic groups. These findings highlight the potential transmission of ARGs through probiotic products in poultry, emphasizing the need for careful selection and monitoring of probiotics concerning antimicrobial resistance traits. Ongoing surveillance and research are crucial to ensure that probiotic benefits do not inadvertently contribute to antibiotic resistance spread.

Further research into the molecular genetics of probiotics is imperative. Future studies should include transcriptomic analyses to better understand the expression and regulation of ARGs within probiotic strains. Such studies are crucial for developing strategies to mitigate the risks associated with resistance gene transfer through probiotics in poultry farming.

## Figures and Tables

**Figure 1 animals-14-01927-f001:**
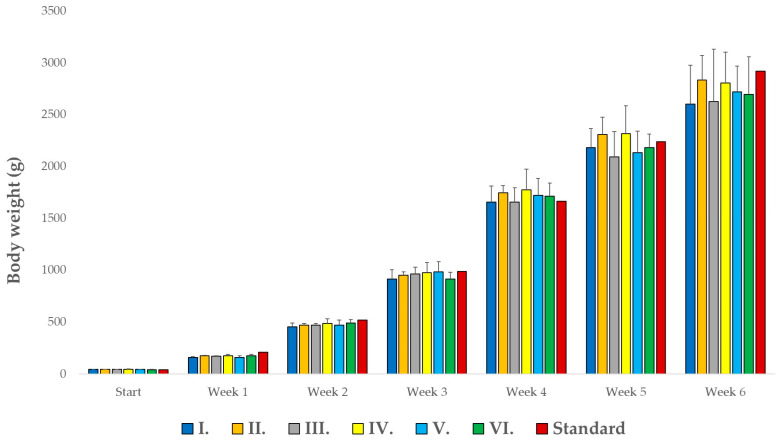
Average weight gain of broilers per week, per group (n = 10 chickens/group). Group I served as the negative control and did not receive probiotics. Group II was treated with BioPlus YC, containing *Bacillus licheniformis* and *Bacillus subtilis*. Group III received Agroferm M+C, which includes *Enterococcus faecium*, *Lactobacillus plantarum*, and *Pediococcus acidilactici*. Group IV was given Ecobiol WX, containing *Bacillus amyloliquefaciens*. Group V received Gastroferm M+C, consisting of *Lactobacillus plantarum*, *Pediococcus acidilactici*, and *Enterococcus faecium*. Finally, Group VI was administered Fecinor Soluble Plus, which contains *Enterococcus faecium*.

**Figure 2 animals-14-01927-f002:**
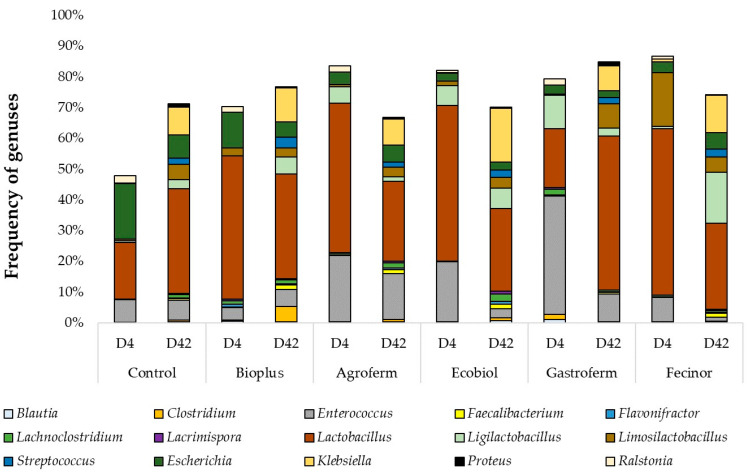
Composition and variation of each sample in response to rearing and different probiotics fed at genus level, expressed as % of total sequence. D—day.

**Figure 3 animals-14-01927-f003:**
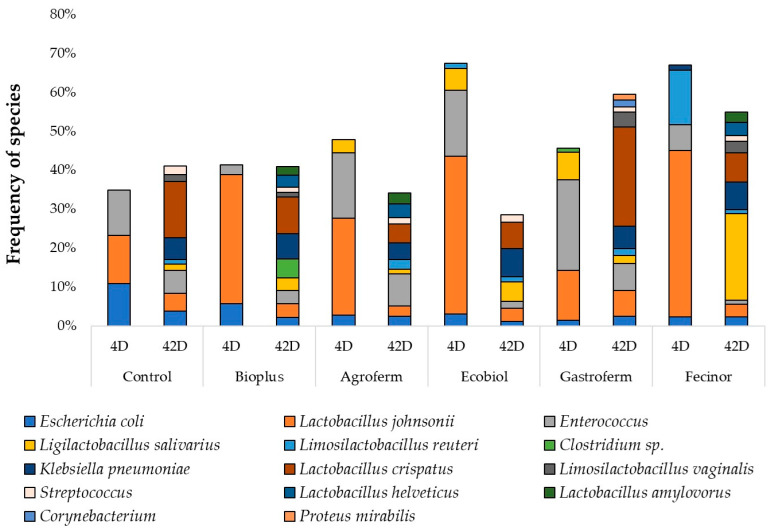
Changes in the microbiome composition at 4 and 42 days of age. D—day.

**Figure 4 animals-14-01927-f004:**
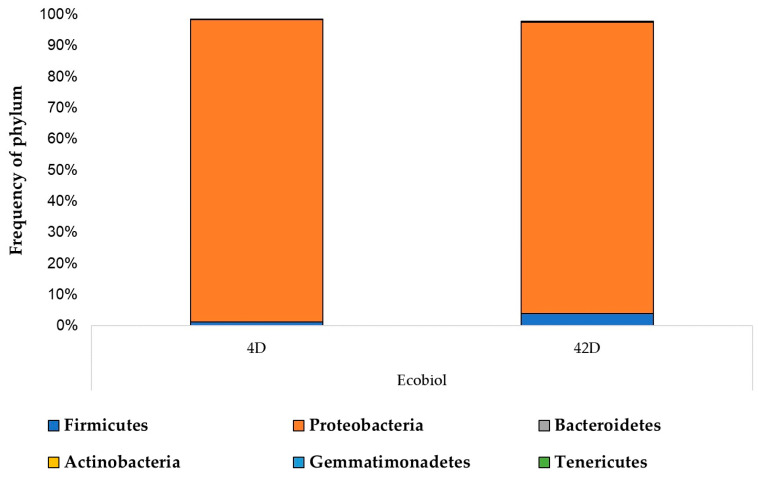
The enrichment solution containing tiamulin and florfenicol not only eliminated resistant strains in the samples but also led to an enrichment of Gram-negative bacteria. D—day.

**Table 1 animals-14-01927-t001:** List and specifications of the selected probiotic products fed mixed with poultry feed.

No.	Product	Probiotic Strain	Tribal Number	CFU/g	Target Animal
1.	BioPlus YC	*Bacillus licheniformis*	DSMZ5749	1.6 × 10^9^	poultry, swine, cattle
		*Bacillus subtilis*	DSMZ5750	1.6 × 10^9^	
2.	Agroferm M+C	*Enterococcus faecium*	DSM7134	1 × 10^9^	broiler, goose, turkey, duck
		*Lactobacillus plantarum*	DSM12837		
		*Pediococcus acidilactici*	DSM16243		
3.	Ecobiol WX	*Bacillus amyloliquefacieus*	CECT5940	1 × 10^10^	poultry
4.	Gastroferm M+C	*Lactobacillus plantarum*	DSM12837	1 × 10^9^	poultry
		*Pediococcus acidilactici*	DSM16243		
		*Enterococcus faecium*	DSM7134		
5.	Fecinor Soluble Plus	*Enterococcus faecium*	CECT4515	1 × 10^10^	poultry, swine

**Table 2 animals-14-01927-t002:** ARGs identified in each formulation. Coverage is the accuracy of the sequencing, with multiple coverage ensuring that the consensus is error-free. Identity is the overlap with the matched sequence.

Product	Sequence %		ARG	Mechanism	Resistance
	Coverage	Identity			
BioPlus YC	100	99.25	*blt*	efflux pump	fluoroquinolones
	100	98.91	*vmlR*	target mutation	macrolides, lincosamides, pleuromutilinek, phenicols, tetracyclines, streptogramin, oxazolidone
	100	98.59	*aadK*	enzymatic inactivation	aminoglycosides
	100	99.49	*tmrB*	permeability reduction	nucleosides
	100	99.79	*lmrB*	efflux pump	lincosamides
	100	98.68	*mphK*	enzymatic inactivation	macrolides
	100	100	*mprF*	target mutation	peptide antibiotics
	100	99.74	*bmr*	efflux pump	fluoroquinolones, phenicols, nucleosides
	86.15	35.52	*dfrA43*	target mutation	diaminopyrimidines
	100	99.65	*ermD*	target mutation	macrolides, lincosamides
	100	98.03	*bcrC*	target mutation	peptide antibiotics
	100	98.56	*bcrB*	efflux pump	peptide antibiotics
	96.08	99.32	*bcrA*	efflux pump	peptide antibiotics
	100	100	*ykkC*	efflux pump	aminoglycosides, tetracyclines, phenicols
	100	99.05	*ykkD*	efflux pump	aminoglycosides, tetracyclines, phenicols
Agroferm M+C	100	100	*efmA*	efflux pump	macrolides, fluoroquinolones
	100	98.9	*AAC(6’)-Ii*	enzymatic inactivation	aminoglycosides
	100	97.15	*msrC*	target mutation	macrolides, lincosamides, pleuromutilinek, phenicols, tetracyclines, streptogramin, oxazolidone
	100	99	*eatAv*	target mutation	
Ecobiol WX	100	99.43	*clbA*	target mutation	phenicols, lincosamides, pleuromutilinek, streptogramin, oxazolidone
	82.56	35.75	*dfrA43*	target mutation	diaminopyrimidines
Gastroferm M+C	100	100	*efmA*	efflux pump	macrolides, fluoroquinolones
	100	97.15	*msrC*	target mutation	macrolides, lincosamides, pleuromutilines, phenicols, tetracyclines, streptogramin, oxazolidone
	100	98.9	*AAC(6’)-Ii*	enzymatic inactivation	aminoglycosides
	100	99	*eatAv*	target mutation	macrolides, lincosamides, pleuromutilines, phenicols, tetracyclines, streptogramin, oxazolidone
Fecinor Soluble Plus	100	98.9	*AAC(6’)-Ii*	enzymatic inactivation	aminoglycosides
	100	97.15	*msrC*	target mutation	macrolides, lincosamides, pleuromutilines, phenicols, tetracyclines, streptogramin, oxazolidone
	100	99	*eatAv*	target mutation	

**Table 3 animals-14-01927-t003:** Minimum inhibitory concentration (MIC) values for strains isolated from each formulation. Green indicates where phenotypic resistance was observed in addition to the antimicrobial resistance gene (ARG) content presumably caused by them.

No.	Probiotic Strain	PEN	AM	AMC	CTR	GEN	OTC	DOX	TIL
		µg/mL							
1.	*Bacillus licheniformis*	0.12	1	2	16	8	8	0.06	0.5
	*Bacillus subtilis*	1	0.5	0.5	0.25	16	0.125	32	0.5
2.	*Enterococcus faecium*	8	1	1	16	32	0.25	0.125	1
	*Lactobacillus plantarum*	1	16	4	16	8	8	4	2
	*Pediococcus acidilactici*	1	16	8	16	4	4	0.5	0.125
3.	*Bacillus amyloliquefacieus*	0.06	1	2	4	32	0.25	0.25	0.5
4.	*Lactobacillus plantarum*	4	2	0.5	16	32	0.5	0.125	4
	*Pediococcus acidilactici*	1	16	4	16	32	32	16	4
	*Enterococcus faecium*	8	1	1	16	32	0.25	0.125	1
5.	*Enterococcus faecium*	4	1	1	16	32	0.25	0.125	4

PEN—penicillin; AM—amoxicillin; AMC—amoxicillin–clavulanic acid; CTR—ceftriaxone; GEN—gentamicin; OTC—oxitetraciklin; DOX—doxycycline; TIL—tilozin.

**Table 4 animals-14-01927-t004:** Minimum inhibitory concentration (MIC) values for strains isolated from each formulation. Green indicates where phenotypic resistance was observed in addition to the antimicrobial resistance gene (ARG) content presumably caused by them (continued).

No.	Probiotic Strain	FLO	CLI	TIA	VAN	GAT	PSA	SUL	TRI
		µg/mL							
1.	*Bacillus licheniformis*	2	32	64	0.25	0.03	4	64	0.25
	*Bacillus subtilis*	2	2	64	0.25	0.03	2	64	0.5
2.	*Enterococcus faecium*	8	4	64	2	0.5	128	128	128
	*Lactobacillus plantarum*	4	0.06	64	32	2	128	128	64
	*Pediococcus acidilactici*	4	0.06	1	32	0.5	128	64	128
3.	*Bacillus amyloliquefacieus*	2	0.5	64	1	0.03	4	128	0.5
4.	*Lactobacillus plantarum*	8	4	64	32	1	128	128	64
	*Pediococcus acidilactici*	8	0.06	64	32	8	128	128	128
	*Enterococcus faecium*	8	4	64	1	2	128	128	128
5.	*Enterococcus faecium*	8	4	64	2	1	128	128	128

FLO—florfenicol; CLI—clindamycin; TIA—tiamulin; VAN—vancomycin; GAT—gatifloxacin; PSA—potential sulphonamide; SUL—sulfamethoxazole; TRI—trimetoprim.

## Data Availability

The datasets used and/or analyzed during the current study are available from the corresponding author on reasonable request. The sequencing files are available at the LINK below.

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
