# Peer review of "Comprehensive Metagenomic Analysis of Veterinary Probiotics in Broiler Chickens"

_animals, 2024, doi:10.3390/ani14131927_

Round 1

Reviewer 1 Report

Comments and Suggestions for Authors

The authors investigate antimicrobial resistance genes (ARGs) in probiotics approved for poultry using next-generation sequencing, highlighting the public health risk associated with ARGs in broiler chickens. They find metagenomic analysis showing no significant changes corresponding to time or diet across groups that they study. The manuscript effectively ties the findings to public health implications. Overall, the manuscript is well-written and adequately demonstrated. Nevertheless, there are areas that could be enhanced to uphold the quality of the research article.

1.     The introduction section lacks a discussion of the current challenges and research gaps in knowledge regarding antimicrobial resistance (AMR) in probiotics.

2.     The font and color of the text in Figures 1-4 should be improved, as they are not clearly visible to readers.

3.     The authors should provide more justification for selecting these specific strains over others, particularly concerning their potential for ARG transfer.

Author Response

Dear Reviewer 1,

Thank you very much for your very thorough work. We have corrected the manuscript according to the attached document, marking the changes and answering the questions raised.

Yours sincerely,
Adam Kerek

Reviewer 2 Report

Comments and Suggestions for Authors

In this study, the effects of veterinary probiotics on the comprehensive metagenome of broiler chickens were investigated and analyzed in detail. But part of the problem remains, as follows

1. The language of the manuscript needs to be further modified.

2.The target animals for BioPlus YC are poultry and whether it may affect the results of broilers.

3. Result 3.4 is redundant and lacks emphasis.

4. The conclusion section is too long and not condensed.

5. The study as a whole does not have a relatively clear conclusion.

Comments on the Quality of English Language

The language of the manuscript needs to be further modified.

Author Response

Dear Reviewer 2,

Thank you very much for your very thorough work. We have corrected the manuscript according to the attached document, marking the changes and answering the questions raised.

Yours sincerely,
Adam Kerek

Reviewer 3 Report

Comments and Suggestions for Authors

The manuscript by Kerek and colleagues focusses on metagenomic (antibiotic resistance genes) analysis of probiotics. The manuscript is interesting and focusses on not explored problem yet. However, the main issue with the manuscript is the lack of details in Material and Methods section and number of samples.

1.       I do not think anybody wants to replace the use of antibiotics with probiotics, rather replace use of antibiotics as growth promoters with probiotics.

2.       There are no conclusions in abstract.

3.       Broiler nurturing diet should be replaced by broiler grower diets.

4.       It is hard to determine, based on the Material and Methods section, what was the replication value for the experiment? Were there 10 birds per probiotic? What was the experimental unit?

5.       There are no details what was sampled, how many samples per treatment.

6.       What ANOVA was used for analysis (one-way or two-way) and why?

7.       NGS needs more details. How were the samples prepared for sequencing? Who sequenced the samples? The number of samples is too low.

8.       The Bioinformatics section needs more details.

9.       Figure 1: what was the n value, how these data were analyzed? This should be analyzed by two-way ANOVA with time and treatment as main factors.

10.   Lines 329-334: how was this data analyzed, what test was used to determine significance?

11.   Lines 341-347: How was this determined. There is nothing in Material and Methods about it.

1 Line 463: The authors should be careful with this statement. They only analyzed feces (excreta) and in birds they are not representative of gut microbiota.

L Line 550: fecal resistome not intestinal resistome.

Comments on the Quality of English Language

N/A

Author Response

Dear Reviewer 3,

Thank you very much for your very thorough work. We have corrected the manuscript according to the attached document, marking the changes and answering the questions raised.

Yours sincerely,
Adam Kerek

Round 2

Reviewer 2 Report

Comments and Suggestions for Authors

The whole manuscript has been modified and complete, and the logic is reasonable.

Comments on the Quality of English Language

Minor editing of English language required.

Reviewer 3 Report

Comments and Suggestions for Authors

Responses to reviewer comments are acceptable. 

Comments on the Quality of English Language

N/A